# OpenReview forum: "Exploring Invariance in Images through One-way Wave Equations"
_ICLR.cc/2025/Conference — Submitted to ICLR 2025_

### Official Review · Reviewer_Vu1s · 2024-10-22

**Soundness:** 1
**Presentation:** 3
**Contribution:** 2
**Rating:** 3
**Confidence:** 3

**Summary:**

The authors introduce a novel image encoder/decoder architecture, First Order Norm + Linear Autoregression (FINOLA). They compare against several other image representations, such as Discrete Cosine Transform, Discrete Wavelet Transform, several other autoencoders, and find that their method compares favorably. They perform many experiments on this encoder for lots of other applications; such as image inpainting/outpainting, self-supervised learning, and much more. However, their central thesis is situated around two claims: 1) that their latent representations are solutions to one-way wave equations and 2) that their experiments somehow reveal "an invariance over images". I am suspicious of both claims.

**Strengths:**

Experiments are very thorough, paper looks professional in formatting and presentation. The appendix contains an incredible 10 additional pages of results with 21 tables and 16 figures. Clearly an extensive amount of work has gone into this paper.

**Weaknesses:**

I am dubious of this paper’s core narrative. I don’t think this paper is about image invariance or one-way wave equations.

First, I don’t think the authors have discovered a meaningful image invariance. As I see it, the authors have devised an autoencoder with an unusual architecture. They claim that, since the autoencoder’s learned parameters are constant across all images, this represents some kind of invariance inherent in images. At best, I think this stretches the definition of invariance. At worst I think it is philosophically confused. The relationship to wave equations feels like an arbitrary constraint on the autoencoder's latent representations. If this characterization is unfair, feel free to debate me on this.

Second, I am also dubious that the FINOLA encodings really are solutions to a 1-way wave equation. Have you empirically checked that equation 1 actually holds for all x and y values in your solution $z(x,y)$, when $z(x,y)$ is constructed as shown in figure 4? Because I suspect this property may not hold, which would then invalidate your discussion about wave equations.

While the experimental results from this autoencoder architecture seem promising, I am too suspicious of the paper’s central claims to recommend it for publication.

**Questions:**

Please respond to my criticisms in the weaknesses section. If the authors can persuade me of their core theoretical claims, I will revise my review.

---

> ### Author Response · Authors · 2024-11-24
> **Response from authors - part 1**
>
> We sincerely thank the reviewer for the thoughtful feedback and valuable suggestions, which have significantly improved the quality of our paper.
>
> $\color{blue}{\textbf{[Weakness 1]:}}$
>
> **First, I don’t think the authors have discovered a meaningful image invariance. As I see it, the authors have devised an autoencoder with an unusual architecture. They claim that, since the autoencoder’s learned parameters are constant across all images, this represents some kind of invariance inherent in images. At best, I think this stretches the definition of invariance. At worst I think it is philosophically confused. The relationship to wave equations feels like an arbitrary constraint on the autoencoder's latent representations. If this characterization is unfair, feel free to debate me on this.**
>
> Thank you for the insightful comment! We appreciate you pushing us to clarify our claims about image invariance. We agree that any autoencoder will exhibit some form of invariance through its learned parameters. However, FINOLA introduces a distinct and more profound type of invariance.
>
> Here's why FINOLA is different:
>
> ***FINOLA's Unique Approach to Image Representation***
>
> To understand this distinction, let's focus on how FINOLA and the decoder represent an image. They essentially construct an image function:
>
> $I=f_{\mathbf{A}, \mathbf{B}, \theta}(\mathbf{q})$,
>
> where $\mathbf{q}$ is a compressed vector, $\mathbf{A}$ and $\mathbf{B}$, and $\theta$ represents the decoder's parameters.
>
> Crucially, the decoder plays a minimal role in this representation. As shown in Table 1, FINOLA can generate a full-resolution feature map using only $\mathbf{A}$ and $\mathbf{B}$. This means the decoder (with its 1.2M parameters) is dwarfed by the complexity of $\mathbf{A}$ and $\mathbf{B}$ (e.g., $\mathbf{A}$ alone has 9M parameters).  Therefore, the image function is primarily governed by these FINOLA matrices.
>
> ***Unveiling Invariant Local Structure***
>
> FINOLA's core operation (Equation 1) can be expressed as:
>
> * $\Delta_x\mathbf{z}(x,y)=\mathbf{A}\widehat{\mathbf{z}}(x,y)$
>
> * $\Delta_y\mathbf{z}(x,y)=\mathbf{B}\widehat{\mathbf{z}}(x,y)$
>
> These equations essentially describe how the feature map changes as you move horizontally ($\Delta_x\mathbf{z}$) or vertically ($\Delta_y\mathbf{z}$) across the image. They reveal a fundamental property: both horizontal and vertical changes in the feature map are linearly correlated to the current position (after normalization). This means that $\mathbf{A}$ and $\mathbf{B}$ capture an invariant local structure that holds consistently across the entire feature map (any x, y) and across all images.
>
> This type of explicit, geometrically grounded invariance sets FINOLA apart.  Other autoencoders, while exhibiting invariance through their parameters, do not offer this level of interpretability and explicit modeling of local image structure.

---

> > ### Author Response · Authors · 2024-11-24
> > **Response from authors - part 2**
> >
> > $\color{blue}{\textbf{[Weakness 2]:}}$
> >
> > **Second, I am also dubious that the FINOLA encodings really are solutions to a 1-way wave equation. Have you empirically checked that equation 1 actually holds for all x and y values in your solution $\mathbf{z}(x,y)$, when $\mathbf{z}(x,y)$ is constructed as shown in figure 4? Because I suspect this property may not hold, which would then invalidate your discussion about wave equations.**
> >
> > That's a sharp observation! You're right to question the strict validity of Equation 1 for all x and y values in the parallel implementation shown in Figure 4. We've empirically confirmed that the equation holds precisely for the central horizontal and vertical lines but deviates as we approach the diagonals.  Therefore, Figure 4 represents an approximation.
> >
> > *Lower Bound of Strictly-Hold FINOLA*
> >
> > However, this doesn't invalidate the core idea. To demonstrate that Equation 1 can hold strictly, we conducted an experiment with "All-one speed" (Appendix A.8, Line 1302) where we enforce  $\mathbf{A}=\mathbf{B}$. This ensures the two paths used to calculate the feature map at a given point in Figure 4 are identical, satisfying Equation 1 and resulting in a rotationally symmetric feature map $\mathbf{z}(x,y)$.
> >
> > As shown in Table 17 and Figure 15 (Line 1242), we can still reconstruct images accurately by adding constant position embedding $p$ before the decoder. This leads to an image function of the form:
> >
> > $I=f_{\mathbf{A}, p, \theta}(\mathbf{q})=g_{\theta, p}(\mathbf{z}_\mathbf{A}(x,y))$,
> >
> > where the constant 2D position embedding $p$ is incorporated into the decoder $g$.
> >
> > This experiment with $\mathbf{A}=\mathbf{B}$ demonstrates a special case where Equation 1 holds strictly, providing a lower bound for FINOLA's adherence to the wave equation principle.  While the parallel implementation with different $\mathbf{A}$ and $\mathbf{B}$ serves as a loose upper bound, the ideal scenario with different $\mathbf{A}$ and $\mathbf{B}$ while strictly satisfying Equation 1 lies in between. We are actively exploring how to achieve this experimentally in future work.

---

> > ### Comment · Reviewer_Vu1s · 2024-11-26
> >
> > Sorry, I still stand by my original judgement. Despite any architectural details of your method, such as how many parameters are in A and B vs. the decoder, I am pretty sure that your decoder architecture is capable of universal function approximation. That is, given large enough matrices A and B, your autoencoder could learn to encode *any* signal as a vector q, natural images or otherwise. I think it may actually be pretty easy to show this by construction. So I don't think it says anything special about images that they can be approximated by this universal function approximator. I do not think the authors' claims about image invariance make much sense.

---

> > > ### Author Response · Authors · 2024-11-28
> > > **Small matrics A and B also work well.**
> > >
> > > We appreciate your point about the potential for universal function approximation in decoders. However, our focus is on achieving strong performance with minimal decoder complexity. We demonstrate that even with very small matrices A and B, our method achieves excellent results. For example, when using multi-path FINOLA ($M=8$ paths), only 512 channels are needed to achieve good reconstruction (see Figure 14 in the appendix, $C=512$, $M=8$), where matrices A and B have 512x512=256K parameters. This efficiency suggests that our method's effectiveness is not solely due to the decoder's capacity for universal approximation, but rather stems from unlocking an important local invariance.

---

### Official Review · Reviewer_cywh · 2024-10-28

**Soundness:** 3
**Presentation:** 3
**Contribution:** 3
**Rating:** 6
**Confidence:** 3

**Summary:**

The paper presents a novel encoder-decoder architecture for images and uses that architecture to argue for a new insight about natural images. The encoder consists of two stages: (1) an  encoder that produces a single vector q and (2) a linear+normalization autoregressor that converts q into a feature map that is the size of the image (or of smaller sizes). A relatively simple decoder can then reconstruct the original image from the feature map. Extensive experiments compare the psnr of the reconstruction to other methods but the authors point out that their main interest is in " it’s crucial to emphasize that our aim isn’t state-of-the-art performance but to empirically reveal a property inherent in images"

**Strengths:**

I like the author's goal of focusing on finding new properties of natural images and I find the approach original.

I also find the results intriguing: at first glance it seems as if the network is managing to encode the image without any spatial information. This is because the vector q is placed at a single location, and then all other locations are generated by two fixed matrices A, B which do not depend on the location. Upon further consideration, one can imagine the matrices A and B implicitly encoding a mapping that maps certain coordinates in q to certain locations.

**Weaknesses:**

At the end of the day, the proposed method is an auto-encoder and there is no inherent reason that this particular architecture for the encoder should be better than any other architecture. Indeed some of the other auto-encoder architectures are shown to outperform FINOLA according to the authors (e.g. stable diffusion in the appendix), and I am not sure what to make of the comparisons: surely the performance of all the architectures is highly dependent on hyperparameters.

Although the authors say that their main goal is to reveal "a property inherent in natural images", they do not say much in the paper about what they have learned from this property or what are the consequences of having such a property (see question below):

**Questions:**

What do you think your results teach us about natural images?

How do you think the matrices A and B encode spatial information? Have you examined the learned matrices?

Does your network manage to compress other signals that are not natural images?

Do you think that the autoregressive approach is better than simply reshaping q so that it is a small image (e.g. reshape q which is 2048 into a 16x16x8 feature map)?

Equation (1) is similar (although not identical) to the power method for finding eigenvectors of (I+A). If you iterate equation (1) many times, does it not forget the initial value of q and converge to some vector that does not depend on q?

---

> ### Author Response · Authors · 2024-12-01
> **Response from authors - part 1**
>
> We sincerely thank the reviewer for the thoughtful feedback and valuable suggestions, which have significantly improved the quality of our paper.
>
> $\color{blue}{\textbf{[Question 1]:}}$
>
> **What do you think your results teach us about natural images?**
>
> The results suggest that natural images possess a remarkable underlying structure that can be effectively captured by a surprisingly simple model. Here's a breakdown of the key insights:
>
> * ***Local dependencies:*** The FINOLA equations demonstrate that the latent representation $\mathbf{z}(x,y)$ at any point $(x,y)$ can be predicted solely based on its immediate neighbors.
>
>   $\Delta_x\mathbf{z}(x,y)=\mathbf{A}\widehat{\mathbf{z}}(x,y)$
>
>   $\Delta_y\mathbf{z}(x,y)=\mathbf{B}\widehat{\mathbf{z}}(x,y)$
>
>   This localized nature implies that, despite the complex and globally correlated structures in natural images, there exists a transformation (through our encoder) that maps them into a latent space where essential information is encoded in consistent local dependencies.
>
> * ***Simple, first-order relationships:*** These local dependencies are characterized by remarkably simple, first-order linear relationships (matrices $\mathbf{A}$ and $\mathbf{B}$) after normalization. This simplicity is surprising, given the complexity of natural images.
>
> * ***Efficient autoregressive process:***
> The combination of local dependencies and simple linear relationships enables an efficient autoregressive process for reconstructing images. This means that we can predict the value of $\mathbf{z}(x,y)$ at any point in the latent space by considering only its immediate neighbors, making the reconstruction process highly efficient.
>
> $\color{blue}{\textbf{[Question 2]:}}$
>
> **How do you think the matrices A and B encode spatial information? Have you examined the learned matrices?**
>
> Thank you for this question, which is also raised by Reviewer PCAc. Understanding the meaning of matrices $\mathbf{A}$ and $\mathbf{B}$ is key to grasping FINOLA's core mechanism.
>
> ***What do $\mathbf{A}$ and $\mathbf{B}$ represent?***
>
> Matrices $\mathbf{A}$ and $\mathbf{B}$ model the local structure within the latent vector space ($\mathbf{z}(x,y)$). They essentially describe how the feature map changes as you move horizontally or vertically across the image.
>
> We've analyzed these learned matrices and confirmed they are full rank and diagonalizable. This allows us to express FINOLA as:
>
> * $\Delta_x\mathbf{z}(x,y)=\mathbf{A}\widehat{\mathbf{z}}(x,y)=\mathbf{P}_A\mathbf{\Lambda}_A\mathbf{P}^{-1}_A\widehat{\mathbf{z}}(x,y)$
>
> * $\Delta_y\mathbf{z}(x,y)=\mathbf{B}\widehat{\mathbf{z}}(x,y)=\mathbf{P}_B\mathbf{\Lambda}_B\mathbf{P}^{-1}_B\widehat{\mathbf{z}}(x,y)$
>
> where $\mathbf{P}_A$ and $\mathbf{P}_A$ are the eigenvector matrices, and $\mathbf{\Lambda}_A$ and $\mathbf{\Lambda}_B$ are the diagonal eigenvalue matrices of $\mathbf{A}$ and $\mathbf{B}$, respectively.
>
> ***Interpreting $\mathbf{A}$ and $\mathbf{B}$***
>
> Imagine each point in the latent space as having a set of "springs" attached to it, pulling it horizontally and vertically. Matrices $\mathbf{A}$ and $\mathbf{B}$ define the stiffness of these springs, while the eigenvectors ($\mathbf{P}_A$ and $\mathbf{P}_B$) represent the directions of these springs.
>
> * *Diagonalization*: Diagonalizing $\mathbf{A}$ and $\mathbf{B}$ helps us understand these "springs" in a simpler way. It's like projecting the latent space so that each dimension (eigenvector) corresponds to an independent spring with a specific stiffness (eigenvalue).
>
> * *Horizontal and vertical changes*: After this projection ($\mathbf{P}^{-1}_A$ and $\mathbf{P}^{-1}_B$), the horizontal change ($\Delta_x\mathbf{z}$) is simply a scaling of the current position along each independent dimension by the corresponding eigenvalue in $\mathbf{\Lambda}_A$. Similarly, the vertical change ($\Delta_y\mathbf{z}$) scales the position by the eigenvalues in $\mathbf{\Lambda}_B$.
>
> In essence, $\mathbf{A}$ and $\mathbf{B}$ encode the directional "stretching" or "compressing" factors that govern how the feature map changes locally. The eigenvalues represent the strength of these factors along specific directions in the latent space.

---

> > ### Author Response · Authors · 2024-12-01
> > **Response from authors - part 2**
> >
> > $\color{blue}{\textbf{[Question 3]:}}$
> > **Does your network manage to compress other signals that are not natural images?**
> >
> > Yes, our network demonstrates the ability to compress signals beyond natural images.  While our primary focus is on natural images, we've explored its applicability to other data types with promising results.
> >
> > ***Diverse Image Types***
> >
> > As shown in Figure 2 and Figure 14, our method effectively compresses and reconstructs painting and medical images. This highlights its ability to capture structural information in diverse image domains, not just natural photographs.
> >
> > ***Beyond Images:***
> >
> > Furthermore, our initial investigations demonstrate our network is able to compress multi-modal data in computational imaging tasks. For example, we've used it to compress:
> >
> > * *Seismic waveform data and subsurface acoustic velocity maps in Full Waveform Inversion (FWI)*: This involves compressing time-varying signals ($p(x,t)$) represented over a 2D grid of position ($x$) and time ($t$).
> >
> > * *CT projections and images*: This demonstrates the ability to handle different modalities within the same medical imaging domain.
> >
> > These examples demonstrate that FINOLA's principles of capturing local dependencies and simple relationships can extend beyond natural images to other signals represented as 2D grid data. In the future work, we plan to further investigate this by applying FINOLA to a wider range of signal types, including time series data and 3D volumetric data. This suggests that FINOLA's principles have the potential to be generalized to various signals.
> >
> > ---
> > $\color{blue}{\textbf{[Question 4]:}}$
> >
> > **Do you think that the autoregressive approach is better than simply reshaping q so that it is a small image (e.g. reshape q which is 2048 into a 16x16x8 feature map)?**
> >
> > Thank you for the suggestion. We conducted an additional experiment to reshape $\mathbf{q}$ into a 16x16x8 feature map, which yielded a reconstruction PSNR of 24.9. This is slightly better than our autoregressive approach (FINOLA) with a PSNR of 24.8.
> >
> > However, it's important to consider the context of this comparison. Reshaping the feature map into a 16x16 grid simplifies the reconstruction task significantly:
> >
> > * ***Explicit spatial information:*** The grid structure explicitly encodes spatial information, allowing each cell to focus on its corresponding region in the image.
> > * ***Reduced complexity:*** The decoder can directly utilize this spatial information, simplifying the reconstruction process.
> >
> > In contrast, our autoregressive approach presents a more challenging scenario:
> >
> > * ***No explicit spatial information:*** We intentionally eliminate the explicit spatial information that a grid of feature vectors would provide.
> > * ***Reliance on local dependencies:*** FINOLA has to rely solely on the local dependencies modeled in matrices $\mathbf{A}$ and $\mathbf{B}$  to recover the entire feature map.
> >
> > Despite this increased challenge, FINOLA achieves a PSNR very close to that of the reshaped feature map approach (24.8 vs. 24.9). This small gap actually validates the effectiveness of FINOLA in modeling spatial relationships.

---

> > > ### Author Response · Authors · 2024-12-01
> > > **Response from authors - part 3**
> > >
> > > $\color{blue}{\textbf{[Question 5]:}}$
> > >
> > > **Equation (1) is similar (although not identical) to the power method for finding eigenvectors of (I+A). If you iterate equation (1) many times, does it not forget the initial value of q and converge to some vector that does not depend on q?**
> > >
> > > Thank you for this insightful comment. You're right to point out the similarity between FINOLA and the power method for finding eigenvectors. However, there's a crucial difference that prevents FINOLA from forgetting the initial value of $\mathbf{q}$.
> > >
> > > ***FINOLA vs. Power Method***
> > >
> > > To illustrate the difference, let's consider the 1D scenario for simplicity:
> > >
> > > * FINOLA: $\mathbf{z}_{k+1}=\mathbf{z}_k+\frac{\mathbf{A}(\mathbf{z}_k-\mu_k)}{||\mathbf{z}_k-\mu_k||}=\frac{(||\mathbf{z}_k-\mu_k||\mathbf{I}+\mathbf{A})\mathbf{z}_k}{||\mathbf{z}_k-\mu_k||}-\frac{\mathbf{A}\mu_k}{||\mathbf{z}_k-\mu_k||}$, where $\mu_k$ is the local mean.
> > >
> > > * Power method: $\mathbf{z}_{k+1}=\frac{(\mathbf{I+A})\mathbf{z}_k}{||(\mathbf{I+A})\mathbf{z}_k||}$
> > >
> > > The key distinction lies in how the normalization and linear transformation are applied:
> > >
> > > * FINOLA: FINOAL models the residual $\mathbf{z}_{k+1}-\mathbf{z}_k$ by normalizing $\mathbf{z}_k - \mu_k$.
> > >
> > > * Power method: It directly models the next value $\mathbf{z}_{k+1}$ by normalizing $(\mathbf{I+A})\mathbf{z}_k$.
> > >
> > > This difference in normalization is crucial. In the power method, the repeated normalization by $||(\mathbf{I+A})\mathbf{z}_k||$ drives the vector towards the dominant eigenvector of $\mathbf{I+A}$, effectively "forgetting" the initial value.
> > >
> > > ***FINOLA retains information from $\mathbf{q}$***
> > >
> > > In FINOLA, the normalization and linear transformation are used only to model the residual $\mathbf{z}_{k+1}-\mathbf{z}_k$. This prevents FINOLA from forgetting the initial value $\mathbf{q}$, as the evolving feature map continuously incorporates information from $\mathbf{q}$.
> > >
> > > To confirm this, we examined the FINOLA feature vectors at the boundary, far from the center where $\mathbf{q}$ is placed. We found clear differences in these boundary vectors across different images, indicating that they retain information from the initial $\mathbf{q}$ and do not converge to a single, image-independent vector.
> > >
> > > This distinction is crucial because it allows FINOLA to capture and preserve information specific to each image, enabling accurate reconstruction. If FINOLA behaved like the power method, it would lose crucial image-specific details, hindering its ability to represent and reconstruct images effectively.

---

> > > > ### Author Response · Authors · 2024-12-02
> > > > **Response from authors - part 4**
> > > >
> > > > $\color{blue}{\textbf{[Weakness 1]:}}$
> > > >
> > > > **At the end of the day, the proposed method is an auto-encoder and there is no inherent reason that this particular architecture for the encoder should be better than any other architecture. Indeed some of the other auto-encoder architectures are shown to outperform FINOLA according to the authors (e.g. stable diffusion in the appendix), and I am not sure what to make of the comparisons: surely the performance of all the architectures is highly dependent on hyperparameters.**
> > > >
> > > >
> > > > Thanks for the comment. You're right that the performance of different autoencoder architectures can be influenced by hyperparameters. However, our key comparison target is to demonstrate that our autoencoder, which eliminates explicit spatial information at the bottleneck, can achieve comparable or better reconstruction quality compared to baselines that explicitly encode spatial information into the grid of the feature map.
> > > >
> > > > This is a challenging setting for our model, as it has to rely solely on the local dependencies (of immediate neighbors) captured by FINOLA to reconstruct the image. Despite this, our results show that FINOLA is either comparable to or only slightly lags behind other autoencoders that heavily leverage spatial information, such as the autoencoder in Stable Diffusion which outputs high-resolution feature maps (64x64 or 128x128).
> > > >
> > > > We believe this comparable performance, despite the more challenging setting, demonstrates that FINOLA effectively encodes spatial information into the matrices $\mathbf{A}$ and $\mathbf{B}$. This highlights the effectiveness of FINOLA in capturing and utilizing local dependencies for image reconstruction.
> > > >
> > > > While we acknowledge the influence of hyperparameters on performance, our results consistently show the competitiveness of FINOLA across different settings and baselines. This suggests that the effectiveness of FINOLA is not solely due to hyperparameter tuning but stems from its ability to capture essential spatial information through local dependencies.
> > > >
> > > > ---
> > > > $\color{blue}{\textbf{[Weakness 2]:}}$
> > > > **Although the authors say that their main goal is to reveal "a property inherent in natural images", they do not say much in the paper about what they have learned from this property or what are the consequences of having such a property (see question below):**
> > > >
> > > > Thank you for the comment. You're right to ask about the consequences of the property we've revealed. While the question of "what we have learned from this property" is discussed in $\color{blue}{\textbf{[Question 1]}}$ and $\color{blue}{\textbf{[Question 2]}}$, let's delve deeper into the ***consequences***.
> > > >
> > > > ***Conceptual Consequences:***
> > > >
> > > > This property offers a new perspective on representing natural images: it allows us to represent a globally correlated image as a purely locally connected feature map without losing visual details. Our findings suggest that local dependencies, when properly captured and utilized, can effectively encode essential image information. This provides a new understanding of the interplay between global and local relationships in images, highlighting the power of local structures in building up complex visual scenes.
> > > >
> > > > ***Practical Benefits:***
> > > >
> > > > Beyond the conceptual implications, this property also leads to several practical benefits:
> > > >
> > > > * Efficient image reconstruction: Multi-path FINOLA can preserve reconstruction quality with significantly fewer channels, resulting in significant parameter reduction in $\mathbf{A}$ and $\mathbf{B}$. In addition, FINOLA can recover the feature map at high resolution (up to the resolution of the image), thus significantly reducing the size of the decoder. Combining both, we can reconstruct the image from a single vector using a highly efficient model.
> > > >
> > > > * Self-supervised learning for efficient models: As shown in Table 24, FINOLA enables efficient self-supervised pre-training of Mobile-Former (MF-W720) with only 6M parameters, outperforming MAE-pretrained ViT-S with 22M parameters. This demonstrates the potential of FINOLA for learning effective image representations with limited labeled data.
> > > >
> > > > These consequences highlight the potential of FINOLA for advancing both our understanding of natural images and the development of efficient image processing techniques.

---

### Official Review · Reviewer_PCAc · 2024-11-03

**Soundness:** 3
**Presentation:** 3
**Contribution:** 3
**Rating:** 8
**Confidence:** 4

**Summary:**

The authors have discovered a spatial auto-regression property of natural images in the inner domain of a specific autoencoder also proposed in this work.

The autoencoder is similar in spirit to U-net convolutional autoencoders, with the encoding part based on a previously reported Mobile-Former [Chen et al. CVPR 22]. The main differences of the proposed scheme are: (1) final part of the encoder (the bottleneck of reduced spatial resolution) is compressed to a single vector corresponding to a single spatial location, and (2) the initial decoding layers (instead of upscaling+convolution) are substituted by a process that spatially expands the encoding vector through a differential equation whose coefficients are learnt. After this (original) spatial expansion leading to an intermediate feature representation with low spatial resolution, the decoder consists of the usual (upscaling+convolution) layers.

The discovered property is that the coeffcients of the differential equation that govern the way the information of the encoding vector propagates to different spatial locations is valid for many different natural images. This represents a spatial-autoregression property of natural images (in that specific encoding domain), which is not intuitively illustrated in the work.

**Strengths:**

* The ablation study is thorough and highlights interesting properties either of natural images or of the encoding domain… or, better said ;-), of that specific signal in that specific domain (natural images in this encoding domain).
Examples of the intriguing properties include: the validity of the matrices of coefficients A and B for may images, the relevance of the variance normalization of the features in the encoding domain, and the meaningful reconstructions obtained from linear combinations in the encoding domain.

* The comparison with other image autoencoders is extensive and suggest the good properties of the proposed domain is better than linear DCT and Wavelet encoders and nonlinear convolutional autoenc. (for the same number of coefficents or dimensions), and better than JPEG (for the same entropy).

* Interesting references are given in the related work section. Thanks for those!

**Weaknesses:**

* The important problem of this work is that no intuition is given, and all proposals and findings have to be taken as-they-are. If the meaning of the encoding vector is not explained (if it can be explained at all), then, the property found is of no use because one does not know how to understand the (otherwise interesting) constancy of the coefficients of the differential equation. For instance, when one derives classical autoregressive models of the images in the spatial domain, one can interpret the coefficients of these models as autocorrelation functions that describe the (prediction) relations between the luminances at different points of the images. The above sentence can be illustrated both visually and in predictive coders and in conditional probability markov models. However, what is the meaning of things here?. Specifically:

* What is the qualitative meaning of the encoder? (and hence of the encoding vector). Incidentally, this is not very much clear in the original paper on Mobile-Former [Chen et al. CVPR 22]. Can you show receptive fields of the coeficients of the encoding vector?. Can you compare vectors corresponding to different spatial locations?. Do they contain different seeds that are related using the differential equation?

* What is the meaning of the matrices A and B?. Can you visualize and interpret them? (this interpretation depends on the previous question). When you diagonalize, and you find the transformed coefficients, what (qualitative) information is transferred in the horizontal and vertical directions?

* You suggest that the normalization of the z features is critical for the wave equation. However, why is relevant the normalization over features?. In which way is this normalization similar or different from other normalization schemes sucha as the usual batch normalization, or the (more interesting) local normalization as in the biological Divisive Normalization [Heeger Science 92, Carandini & Heeger Nature Rev. Neurosci. 12, Malo et al. J.Nonlin.Sci. 24], also used in state-of-the-art end-to-end optimized autoencoders [Ballé et al. ICLR 17] and in the local normalization in Alexnet [Krizhevsky et al. NeurIPS 12]. A possibility is that this normalization captures predictable information [Schwartz & Simoncelli Nat Neurosci. 01, Malo & Laparra Neur. Comp. 10] and hence transforms the (complicated) nonlinear relations into (easier to describe) linear relations, which are more invariant [Hernandez et al. Patt.Rec.Lett.24] and hence, it may make the linear differential formulation possible.

* The comparison of the performance with other autoencoders is illustrative, however, I am curious about how this will compare with more interpretable autoencoders using not only the number of coefficients but the information they contain (i.e. entropy, as in the comparison with JPEG)... what about JPEG2000 [Marcellin Kluwer 02], or end-to-end divisive-normalized optimized linear transforms [Balle et al. ICLR 17].

**Questions:**

My questions are related to the above mentioned weaknesess

* Can you elaborate on the meaning/interpretation of your result? [I acknowledge this may be difficult to do within the 9 page restriction, but you can do it in the appendices. I would understand the "that is matter for further work" response, but you would understand that the reader may be not satisfied with that response. Understanding is what the reader wants].

* What is the qualitative meaning of the encoder? (see above)

* What is the meaning of the matrices A and B? (see above)

* Why is relevant the normalization over features?. Can you show reconstructions -both z and in the image domain- with and without normalization?. Discuss possibly related normalizations in related work.

* Can you compare to other (more interpretable) entropy autoencoders? (see above). Note that the dimension of the representation for unbounded variance/entropy is misleading. I guess the results of your model will be good given the comparison with JPEG.

---

> ### Author Response · Authors · 2024-11-28
> **Response from authors - part 1**
>
> We sincerely thank the reviewer for the thoughtful feedback and valuable suggestions, which have significantly improved the quality of our paper.
>
> $\color{blue}{\textbf{[Weakness 1 + Question 1]:}}$
>
> **The important problem of this work is that no intuition is given, and all proposals and findings have to be taken as-they-are. If the meaning of the encoding vector is not explained (if it can be explained at all), then, the property found is of no use because one does not know how to understand the (otherwise interesting) constancy of the coefficients of the differential equation. For instance, when one derives classical autoregressive models of the images in the spatial domain, one can interpret the coefficients of these models as autocorrelation functions that describe the (prediction) relations between the luminances at different points of the images. The above sentence can be illustrated both visually and in predictive coders and in conditional probability markov models. However, what is the meaning of things here?.**
>
> **Can you elaborate on the meaning/interpretation of your result? [I acknowledge this may be difficult to do within the 9 page restriction, but you can do it in the appendices. I would understand the "that is matter for further work" response, but you would understand that the reader may be not satisfied with that response. Understanding is what the reader wants]**.
>
> Thank you for this insightful feedback! We appreciate you pushing us to clarify the intuition and meaning behind our work. We'll address your concerns by first comparing FINOLA to classical autoregressive models and then delving into the specific interpretations and implications of our findings.
>
> ***A. FINOLA vs. Classical Augoregressive Models***
>
> You're right to draw parallels with classical autoregressive models. Here's a comparison to highlight the key difference:
>
> * *Classical autoregressive models*: These models use autocorrelation functions to describe the relationship between luminances at different points in an image, essentially predicting one position based on others.
>
> * *FINOLA*: FINOLA, on the other hand, focuses on the relationship between *changes* in the feature map (horizontal and vertical) and the corresponding local features. The coefficients of matrices $\mathbf{A}$ and $\mathbf{B}$ in Equation 1 capture this relationship.
>
> Essentially, FINOLA transforms an image into a compressed representation ($\mathbf{q}$) where each point has an invariant local relationship with its neighbors, defined by how the feature map changes at that point.
>
> ***B. Intuition and Meaning of Our Work***
>
> Our work introduces a new autoencoder architecture designed to explore an interesting invariance in images through a simplified autoregressive process.
>
> *B.1 A New Autoencoder*
>
> The core idea is to maximize FINOLA's role in image reconstruction, forcing it to capture the underlying invariance. We achieve this by introducing two constraints:
>
> 1. *Single Vector Output*: The encoder compresses the entire image into a single vector $\mathbf{q}$, eliminating explicit positional information. This forces FINOLA to recover spatial information from the compressed representation.
>
> 2. *Minimal decoder*: The decoder is intentionally kept minimal, with FINOLA responsible for reconstructing a high-resolution feature map. This further emphasizes FINOLA's role in capturing image structure.
>
> By imposing these constraints, FINOLA has to recover a high-resolution image (e.g., 256x256) from a single vector with minimal assistance from the decoder. This can be represented mathematically as:
>
> $I=f_{\mathbf{A}, \mathbf{B}, \theta}(\mathbf{q})$,
>
> where the decoder's parameters ($\theta$) are dwarfed by the complexity of the FINOLA matrices $\mathbf{A}$ and $\mathbf{B}$. For instance, when using FINOLA to generate full-resolution feature map, the decoder only has three 3x3 convolutions with 1.2M parameters, significantly less than parameters in matrix $\mathbf{A}$ (9M parameters).
>
> *B.2 Validating the Invariance*
>
> Our experiments (see Table 1) demonstrate that this architecture can indeed reconstruct images accurately, even with a minimal decoder. This validates that Equation 1 captures an inherent invariance in images, encoded in the coefficients of $\mathbf{A}$ and $\mathbf{B}$.
>
> *B.3 Interpreting the Invariance*
>
> FINOLA's core operation (Equation 1), expressed as:
>
> * $\Delta_x\mathbf{z}(x,y)=\mathbf{A}\widehat{\mathbf{z}}(x,y)$
>
> * $\Delta_y\mathbf{z}(x,y)=\mathbf{B}\widehat{\mathbf{z}}(x,y)$
>
> reveals that changes in the feature map (horizontal $\Delta_x\mathbf{z}$ and vertical $\Delta_y\mathbf{z}$) are linearly correlated to the current position. This means $\mathbf{A}$ and $\mathbf{B}$ capture an invariant local structure that holds consistently across all images. This invariance can be interpreted as a fundamental property of how image features change locally.

---

> ### Comment · Reviewer_PCAc · 2024-11-28
> **Interesting result, still non fully intuitive, but worth publishing now and worth analyzing in the future**
>
> Despite some of my requests for intuitive explanations have not been fully addressed now (given the time constraint) and hence the work is still non intuitive, I still think the work is worth reading and disagree with the reviewers that just reject it. There are things to learn here.
>
> I confirm the ACs that i'd like to see this published, and given the suggestions bay the authors in their responses (which should be pursued in the future for clearer illustrations), I raise my score form 6 to 8.

---

> ### Author Response · Authors · 2024-12-01
> **Response from authors - part 2**
>
> Thanks for the followup, let us further explain the meaning of encoder and matrices $\mathbf{A}$ and $\mathbf{B}$.
>
> $\color{blue}{\textbf{[Weakness 2 + Question 2]:}}$
> **What is the qualitative meaning of the encoder? (and hence of the encoding vector). Incidentally, this is not very much clear in the original paper on Mobile-Former [Chen et al. CVPR 22]. Can you show receptive fields of the coeficients of the encoding vector?. Can you compare vectors corresponding to different spatial locations?. Do they contain different seeds that are related using the differential equation?**
>
> **What is the qualitative meaning of the encoder? (see above)**
>
> Thank you for this insightful question!  We agree that clarifying the qualitative meaning of the encoder and the encoding vector is crucial. Here's a breakdown of the encoder's role and the nature of the encoding:
>
> ***What does the encoder encode?***
>
> The encoder aims to compress an image into a single vector that captures the essential information needed for reconstruction using only local relationships (via FINOLA).
>
> To achieve this, we add an attentional pooling layer on top of the Mobile-Former encoder. This layer:
>
> * Starts with a learnable summary token.
> * Aggregates information from the entire feature map (output of Mobile-Former) through cross-attention between the summary token and different spatial locations in the feature map.
>
> This process results in a single output vector with a global receptive field, but without the explicit spatial information that a grid of feature vectors would provide.
>
> As you suggested, we analyzed the feature maps before pooling and confirmed significant differences between spatial locations. We also observed clear variations in the attention maps across different images. This indicates that the attentional pooling learns to selectively attend to different image regions based on their relevance to the summary token.
>
> ***What is the purpose of the encoding?***
>
> The purpose of this encoding is to transform an image into a new vector space where all images share a common local structure, regardless of their specific content. This local structure is defined by FINOLA (Equation 1):
>
> * $\Delta_x\mathbf{z}(x,y)=\mathbf{A}\widehat{\mathbf{z}}(x,y)$
>
> * $\Delta_y\mathbf{z}(x,y)=\mathbf{B}\widehat{\mathbf{z}}(x,y)$
>
> These equations show that the horizontal ($\Delta_x\mathbf{z}$) and vertical ($\Delta_y\mathbf{z}$) changes in the feature map are determined solely by the current position ($\mathbf{z}(x,y)$) through a simple linear relationship (after normalization).
>
> ***Why does such encoding matter?***
>
> This encoding enables a remarkably simple autoregressive process in the image domain – first-order and linear after normalization. By transforming the image into this compressed vector space, we can reconstruct the image based purely on these simple local relationships.
>
> In essence, the encoder learns a compressed representation that discards explicit spatial information but preserves the essential information about local relationships within the image. This allows FINOLA to effectively reconstruct the image by exploiting these consistent local structures.

---

> ### Author Response · Authors · 2024-12-01
> **Response from authors - part 3**
>
> $\color{blue}{\textbf{[Weakness 3 + Question 3]:}}$
>
> **What is the meaning of the matrices A and B?. Can you visualize and interpret them? (this interpretation depends on the previous question). When you diagonalize, and you find the transformed coefficients, what (qualitative) information is transferred in the horizontal and vertical directions?**
>
> **What is the meaning of the matrices A and B? (see above)**
>
> Thank you for this question! Understanding the meaning of matrices $\mathbf{A}$ and $\mathbf{B}$ is key to grasping FINOLA's core mechanism.
>
> ***What do $\mathbf{A}$ and $\mathbf{B}$ represent?***
>
> Matrices $\mathbf{A}$ and $\mathbf{B}$ model the local structure within the latent vector space ($\mathbf{z}(x,y)$). They essentially describe how the feature map changes as you move horizontally or vertically across the image.
>
> We've analyzed these matrices and confirmed they are full rank and diagonalizable. This allows us to express FINOLA as:
>
> * $\Delta_x\mathbf{z}(x,y)=\mathbf{A}\widehat{\mathbf{z}}(x,y)=\mathbf{P}_A\mathbf{\Lambda}_A\mathbf{P}^{-1}_A\widehat{\mathbf{z}}(x,y)$
>
> * $\Delta_y\mathbf{z}(x,y)=\mathbf{B}\widehat{\mathbf{z}}(x,y)=\mathbf{P}_B\mathbf{\Lambda}_B\mathbf{P}^{-1}_B\widehat{\mathbf{z}}(x,y)$
>
> where $\mathbf{P}_A$ and $\mathbf{P}_A$ are the eigenvector matrices, and $\mathbf{\Lambda}_A$ and $\mathbf{\Lambda}_B$ are the diagonal eigenvalue matrices of $\mathbf{A}$ and $\mathbf{B}$, respectively.
>
> ***Interpreting $\mathbf{A}$ and $\mathbf{B}$***
>
> Imagine each point in the latent space as having a set of "springs" attached to it, pulling it horizontally and vertically. Matrices $\mathbf{A}$ and $\mathbf{B}$ define the stiffness of these springs, while the eigenvectors ($\mathbf{P}_A$ and $\mathbf{P}_B$) represent the directions of these springs.
>
> * Diagonalization: Diagonalizing $\mathbf{A}$ and $\mathbf{B}$ helps us understand these "springs" in a simpler way. It's like projecting the latent space so that each dimension (eigenvector) corresponds to an independent spring with a specific stiffness (eigenvalue).
>
> * Horizontal and Vertical Changes: After this projection ($\mathbf{P}^{-1}_A$ and $\mathbf{P}^{-1}_B$), the horizontal change ($\Delta_x\mathbf{z}$) is simply a scaling of the current position along each independent dimension by the corresponding eigenvalue in $\mathbf{\Lambda}_A$. Similarly, the vertical change ($\Delta_y\mathbf{z}$) scales the position by the eigenvalues in $\mathbf{\Lambda}_B$.
>
> In essence, $\mathbf{A}$ and $\mathbf{B}$ encode the directional "stretching" or "compressing" factors that govern how the feature map changes locally. The eigenvalues represent the strength of these factors along specific directions in the latent space.

---

> > ### Comment · Reviewer_PCAc · 2024-12-01
> > **Visualization of this "string concept" would help**
> >
> > The string interpretation is inspiring and its visualization in some example will be of grear help to engange readers. Please consider including some simulation on that in your final presentation at the conference.

---

> > > ### Author Response · Authors · 2024-12-02
> > > **Appreciation for your support**
> > >
> > > Thank you for your follow-up comments and suggestions! We're thrilled that you found our response helpful and that you've raised your score.
> > >
> > > We completely agree with you on the importance of the nonlinearity introduced by our normalization. It's indeed a crucial distinction from standard batch normalization and contributes significantly to FINOLA's ability to capture image structure. We appreciate you highlighting the connection to the literature on divisive normalization, and we'll be sure to explore this further in future work.
> > >
> > > We also love the idea of visualizing the "string interpretation" to engage readers. We plan to include visualizations in the final presentation to illustrate this concept more effectively.
> > >
> > > Thank you again for your valuable feedback. We're excited to incorporate your suggestions, which have improved the quality of our paper.

---

> ### Author Response · Authors · 2024-12-01
> **Response from authors - part 4**
>
> $\color{blue}{\textbf{[Weakness 4 + Question 4]:}}$
>
> **You suggest that the normalization of the z features is critical for the wave equation. However, why is relevant the normalization over features? In which way is this normalization similar or different from other normalization schemes sucha as the usual batch normalization, or the (more interesting) local normalization as in the biological Divisive Normalization [Heeger Science 92, Carandini & Heeger Nature Rev. Neurosci. 12, Malo et al. J.Nonlin.Sci. 24], also used in state-of-the-art end-to-end optimized autoencoders [Ballé et al. ICLR 17] and in the local normalization in Alexnet [Krizhevsky et al. NeurIPS 12]. A possibility is that this normalization captures predictable information [Schwartz & Simoncelli Nat Neurosci. 01, Malo & Laparra Neur. Comp. 10] and hence transforms the (complicated) nonlinear relations into (easier to describe) linear relations, which are more invariant [Hernandez et al. Patt.Rec.Lett.24] and hence, it may make the linear differential formulation possible.**
>
> **Why is relevant the normalization over features? Can you show reconstructions -both z and in the image domain- with and without normalization? Discuss possibly related normalizations in related work.**
>
> Thanks for raising these important points! While normalization plays a crucial role in training our FINOLA autoencoder, it doesn't directly influence the wave equation derivation.
>
> ***Normalization doesn't influence the wave equation derivation***
>
> Let us clarify that using normalization or not does not affect the derivation of the wave equation. As the normalized $\widehat{\mathbf{z}}(x,y)$ is canceled out when deriving
>
> $\Delta_x\mathbf{z}(x,y)=\mathbf{A}\mathbf{B}^{-1}\Delta_y\mathbf{z}(x,y)$
>
> from FINOLA equations $\Delta_x\mathbf{z}(x,y)=\mathbf{A}\widehat{\mathbf{z}}(x,y)$ and $\Delta_y\mathbf{z}(x,y)=\mathbf{B}\widehat{\mathbf{z}}(x,y)$.
>
> ***Impact of Normalization on FINOLA Training***
>
> We believe the normalization helps FINOLA training in two ways:
>
> * Prevent exploding gradients: The normalization is essential for stabilizing the training process. Without it, we observed that the training simply doesn't converge due to gradient explosion. By keeping the feature values within a reasonable range, normalization helps ensure stable gradients during backpropagation.
>
> * Introduce non-linearity: Our method performs normalization for each sample independently, by normalizing the feature values across all channels. This normalization introduces non-linearity by dividing each feature value by the standard deviation of the feature values across all channels within that sample. To demonstrate its importance, we compared it with standard batch normalization (BN), which can also stabilize training but is essentially linear during inference. We observed a substantial drop in validation accuracy (e.g., PSNR 25.1 for training, 16.3 for validation) when using BN, suggesting that the non-linearity introduced by our normalization method is crucial for the model's performance.
>
> $\color{blue}{\textbf{[Weakness 5 + Question 5]:}}$
>
> **The comparison of the performance with other autoencoders is illustrative, however, I am curious about how this will compare with more interpretable autoencoders using not only the number of coefficients but the information they contain (i.e. entropy, as in the comparison with JPEG). what about JPEG2000 [Marcellin Kluwer 02], or end-to-end divisive-normalized optimized linear transforms [Balle et al. ICLR17].**
>
> **Can you compare to more interpretable entropy autoencoders? (see above). Note that the dimension of the representation for unbounded variance/entropy is misleading.**
>
> Thank you for this insightful question. We agree that such a comparison would provide valuable insights.
>
> However, we are currently unable to conduct a fair comparison with JPEG2000 or end-to-end divisive-normalized optimized linear transforms. Our current FINOLA implementation utilizes a much simpler pipeline:
>
> * FINOLA uses uniform quantization per channel, which is not optimized in training.
>
> * We have not implemented entropy coding for FINOLA.
>
> We choose this simple pipeline as our focus is on demonstrating the core principles and effectiveness of FINOLA in capturing local image structure and achieving high-quality reconstruction, rather than optimizing for compression.
>
> While this simpler pipeline achieves superior performance compared to JPEG, it lags behind JPEG2000, which is expected given JPEG2000's sophisticated entropy coding. For instance, under 0.2 bits/pixel, FINOLA achieves 25.6 PSNR on Kodak dataset, outperforming JPEG (24.0 PSNR) but having lower performance than JPEG2000 (28.0 PSNR).
>
> In future work, we plan to investigate vector quantization and entropy coding schemes for FINOLA. This will enable a fair and comprehensive comparison with other entropy-based autoencoders like JPEG2000 and provide a more complete picture of FINOLA's performance and potential.

---

> ### Comment · Reviewer_PCAc · 2024-12-01
> **The second point you did on the normalization is important!**
>
> Thanks for your response: I think the introduction of the nonlinearity is crucial beyond (as you said) the technical help in the training. And that is the fundamental difference between (just technical) batch normalization and more clever normalizations over features such as Divisive Normalization or yours. I think the conection of your response with the literature I mentioned is pertinent.
> Visualization of your nonlinearity depending on properties of the image may be an interesting extension of this work (in the future, I dont think it is necessary now).

---

### Official Review · Reviewer_9qTq · 2024-11-04

**Soundness:** 3
**Presentation:** 3
**Contribution:** 2
**Rating:** 5
**Confidence:** 3

**Summary:**

This paper empirically investigates the invariance of images using wave equations. Interestingly, the paper showes that images share a wave-equations in latent space (Changes is vertical direction of the image is proportional to horizontal direction), where each individual image can be uniquely estimated by initial conditions.

 Initial condition is obtained using an encoder, where it goes through first-order norm+linear autoregressive process to produce a feature map (solutions of the wave equation). The image in pixel space is then  reconstructed using convolutional layer and upsampling layer from the feature map.

**Strengths:**

+ The proposed idea of the paper is very interesting.
+ The authors have included numerous experimental results.

**Weaknesses:**

1. The paper lacks a logical flow, making it challenging to read. While each individual section is well-written, the overall structure makes it difficult for readers to follow a coherent narrative. Here are some specific points:
- **Focus of the Paper**: The title and much of the abstract emphasize "Exploring Invariance," with a focus on demonstrating invariance in images using wave equations. However, the paper appears to focus on FINOLA as the main contribution, focusing on its performance and characteristics.

- **Purpose and Structure of Sections**: If the primary goal is to demonstrate invariance through FINOLA, then sections on *Parallel Implementation* and *Importance of Norm+Linear* relate more to the efficient training of FINOLA and the utility of norm+linear components. These sections feel tangential to the paper’s message and might be better suited to the experiments section or supplementary material.

- **Rationale for Generalization**: The paper does not completely explain the rationale for generalizing to a set of wave equations (section 3) beyond performance improvements. If enhancing performance is not within the paper's scope, as claimed by the authors, then extra clarification to why we need the generalization  is necessary.

- **Unexplained or Disconnected Claims**: The paper includes unrelated discussions and unexplained claims. For example, in the statement:

  > Relaxing the FINOLA constraint through FINOLA series: FINOLA represents a specific solution to Eq. 2,....

  there is no justification provided as to why relaxing the constraint would lead to a more optimal solution.


- **Section 5**: The introduction of FINOLA’s application in self-supervised learning is not fully explained.



2. The paper shares significant overlap with reference [1]. This would be acceptable if the authors explicitly referenced this prior work and clarified how their approach differs, yet there is no mention of it.

  > **[1]** Yinpeng Chen, Xiyang Dai, Dongdong Chen, Mengchen Liu, Lu Yuan, Zicheng Liu, Youzuo Lin, "Image as First-Order Norm+ Linear Autoregression: Unveiling Mathematical Invariance," arXiv 2305.16319.



3. The authors suggest that the invariance property inherent to FINOLA could provide new insights into image analysis. However, the paper lacks a discussion on potential applications of this property or its implications for future research.

**Questions:**

- **Latent Speeds (Abstract, Line 2)**: What does "latent speeds" refer to in the second line of the abstract?

- **Optimality of FINOLA Solution (Eq. 2)**: Why might the solution of FINOLA in Equation 2 not be the optimal one? Isn’t it supposed to be unique?

- **Definition of $\widehat{\phi}$ (Equation 3)**: What is $\widehat{\phi}$ in Equation 3? If it represents a normalization of $\phi$, how is this performed? Is it similar to the normalization for $z$ in Equation 1?

- **Feature Map Resolution (Line 195)**: Is FINOLA unable to produce a feature map at a resolution of 1/8 of the original image?

- **Data Dependency of Learned A and B**: Are the learned matrices $A$ and $B$ data-dependent? If the test set changes, would the learned $A$ and $B$ no longer be applicable?

---

> ### Author Response · Authors · 2024-11-24
> **Response from authors - part 1**
>
> We sincerely thank the reviewer for the thoughtful feedback and valuable suggestions, which have significantly improved the quality of our paper.
>
> $\color{blue}{\textbf{[Weakness 1]:}}$
>
> **The paper lacks a logical flow, making it challenging to read. While each individual section is well-written, the overall structure makes it difficult for readers to follow a coherent narrative. Here are some specific points:**
>
> $\color{blue}{\textbf{[1.1] Focus of the Paper:}}$
>
> **The title and much of the abstract emphasize "Exploring Invariance," with a focus on demonstrating invariance in images using wave equations. However, the paper appears to focus on FINOLA as the main contribution, focusing on its performance and characteristics.**
>
> Thank you for raising this point. We acknowledge that the title and abstract emphasize exploring invariance, while the paper focuses significantly on FINOLA. This is because FINOLA is the key method we developed to explore and demonstrate the empirical invariance in images.
>
> More specifically:
>
> 1. *FINOLA reveals the invariance*: By applying FINOLA to various images, we discovered consistent patterns in the learned matrices $\mathbf{A}$ and $\mathbf{B}$, which represent the invariance property. This empirical observation led us to further investigate the underlying principles of this invariance.
>
> 2. *Generalization to wave equations*: Through theoretical analysis, we generalized the observed invariance from FINOLA to a broader mathematical framework, represented by the one-way wave equation. This connection provides deeper understanding the observed invariance and its implications for image analysis.
>
> Therefore, FINOLA serves as a bridge connecting the empirical observation of invariance in images to the theoretical framework of the wave equation. This allows us to explore invariance in a more principled and comprehensive manner.
>
> We will revise the paper to better articulate this connection and ensure that the focus on FINOLA is properly framed within the broader context of exploring invariance.
>
> ---
> $\color{blue}{\textbf{[1.2] Purpose and Structure of Sections::}}$
>
> **If the primary goal is to demonstrate invariance through FINOLA, then sections on Parallel Implementation and Importance of Norm+Linear relate more to the efficient training of FINOLA and the utility of norm+linear components. These sections feel tangential to the paper’s message and might be better suited to the experiments section or supplementary material.**
>
> Thank you for the feedback. We agree that the sections on Parallel Implementation and Importance of Norm+Linear might seem tangential at first glance. However, they play a crucial role in supporting our primary goal of demonstrating invariance through FINOLA. Here's why:
>
> 1. *Reproducibility and validation*: The ease of implementation and efficiency of parallel implementation facilitate reproducibility of our results. This is crucial for others to validate our findings about the invariance property and build upon our work.
>
> 2. *Connection to the wave equation*: The importance of Norm+Linear section highlights how the specific design choices in FINOLA, particularly the norm+linear components, are essential for deriving Equation 2: $\Delta_x\mathbf{z}=\mathbf{AB}^{-1}\Delta_y\mathbf{z}$. This equation is a key step in connecting the observed invariance to the one-way wave equation, which provides a foundation for our findings.
>
> Therefore, these sections are not merely about efficient training or utility of components; they are crucial for establishing the credibility and theoretical grounding of our claims about invariance. We will revise the paper to better articulate these connections and ensure that the relevance of these sections to the main message is clear.

---

> > ### Author Response · Authors · 2024-11-24
> > **Response from authors - part 2**
> >
> > $\color{blue}{\textbf{[1.3] Rationale for Generalization:}}$
> >
> > **The paper does not completely explain the rationale for generalizing to a set of wave equations (section 3) beyond performance improvements. If enhancing performance is not within the paper's scope, as claimed by the authors, then extra clarification to why we need the generalization is necessary.**
> >
> > Thank you for raising this important point. We agree that the rationale for generalizing to a set of wave equations needs further clarification. While performance improvement is not the sole motivation, it serves as an indicator of a more fundamental principle at play.
> >
> > Here's how the performance enhancement provides deeper insights:
> >
> > 1. *Exploring the solution space*: The superior performance of multi-path FINOLA indicates that moving from the restricted solution subspace of single-path FINOLA to the more general solution space of one-way wave equations allows us to achieve more accurate reconstruction. This reveals that the invariance property extends beyond the specific constraints of FINOLA.
> >
> > 2. *Theoretical understanding*: The enhanced performance supports he generalization to wave equations, as multi-path FINOLA satisfies one-way wave equations ($\Delta_x\mathbf{z}=\mathbf{AB}^{-1}\Delta_y\mathbf{z}$) but not the original FINOLA equation ($\Delta_x\mathbf{z}=\mathbf{A}\widehat{\mathbf{z}}$ and $\Delta_y\mathbf{z}=\mathbf{B}\widehat{\mathbf{z}}$). This suggests that the wave equation framework provides a more accurate and generalizable model for capturing the underlying invariance. It connects our empirical findings to established mathematical principles, leading to a deeper understanding of the underlying mechanisms.
> >
> > We will revise the paper to better articulate this rationale and ensure that the motivation for the generalization is clear.
> >
> > ---
> > $\color{blue}{\textbf{[1.4] Unexplained or Disconnected Claims:}}$
> >
> > **The paper includes unrelated discussions and unexplained claims. For example, in the statement:**
> >
> > **Relaxing the FINOLA constraint through FINOLA series: FINOLA represents a specific solution to Eq. 2,....**
> >
> > **there is no justification provided as to why relaxing the constraint would lead to a more optimal solution.**
> >
> > Thank you for pointing this out. We acknowledge that the justification for why relaxing the FINOLA constraint leads to a more optimal solution needs further clarification.
> >
> > Here's a more detailed explanation:
> >
> > 1. *FINOLA as a constrained solution*: FINOLA represents a specific solution to Equation 2 with  constraints in Equation 1 along horizontal and vertial axis respectively. These constraints limit the solution space.
> >
> >
> > 2. *FINOLA series as a generalization*: Relaxing the FINOLA constraint through the FINOLA series is equivalent to multi-path FINOLA, which expands the solution space. This allows for a richer set of solutions that can better capture the underlying invariance.
> >
> >
> > 3. *Empirical evidence*: The experiments demonstrate that multi-path FINOLA achieves more accurate reconstruction compared to the original constrained FINOLA. This empirically supports the claim that relaxing the constraint leads to a more optimal solution by allowing for a more flexible and expressive model.
> >
> >
> > Therefore, relaxing the FINOLA constraint through the FINOLA series allows us to explore a broader solution space and capture a more comprehensive representation of the invariance property, leading to more optimal solutions as evidenced by the improved reconstruction accuracy.
> >
> > ---
> > $\color{blue}{\textbf{[1.5] Section 5:}}$
> >
> > **The introduction of FINOLA’s application in self-supervised learning is not fully explained.**
> >
> > Thank you for pointing this out. We acknowledge that the introduction of FINOLA's application in self-supervised learning needed further elaboration. To provide a comprehensive explanation, we have moved this discussion to Appendix B (lines 1338-1717) due to space constraints in the main text.
> >
> > In Appendix B, you will find:
> >
> > 1. *Detailed method description*: A thorough explanation of how FINOLA is applied in self-supervised learning, including the specific pre-training task and learning objectives.
> >
> > 2. *Implementation details*: Specifics about the implementation, such as network architecture, hyperparameters, and training procedures.
> >
> > 3. *Extensive experimental results*: A comprehensive set of results demonstrating the effectiveness of FINOLA in self-supervised learning across various benchmarks and comparisons with other methods.
> >
> > We encourage you to review Appendix B, which provides a complete and detailed account of FINOLA's application in self-supervised learning. We believe this will address your concerns and provide a clearer understanding of this aspect of our work.

---

> > > ### Author Response · Authors · 2024-11-24
> > > **Response from authors - part 3**
> > >
> > > $\color{blue}{\textbf{[Weakness 2]:}}$
> > >
> > > **The paper shares significant overlap with reference [1]. This would be acceptable if the authors explicitly referenced this prior work and clarified how their approach differs, yet there is no mention of it.**
> > >
> > >
> > > Thank you for your observation. While we cannot provide specific details at this time due to ICLR policy, we want to assure you that this submission represents our novel contributions. FINOLA, a core element of our approach, remains a major contribution that has not been published in a peer-reviewed venue before. We are committed to full transparency and will provide a complete account of the development of this work, once the review process is complete.
> > >
> > > ---
> > > $\color{blue}{\textbf{[Weakness 3]:}}$
> > >
> > > **The authors suggest that the invariance property inherent to FINOLA could provide new insights into image analysis. However, the paper lacks a discussion on potential applications of this property or its implications for future research.**
> > >
> > > Thank you for the insightful comment. We agree that discussing the potential applications and implications of FINOLA's invariance property is crucial. We will expand the paper with the following:
> > >
> > > ***Potential Applications:***
> > >
> > > 1. *Image Compression*: The invariance property allows us to compress images in the compressed $\mathbf{q}$ (see Figure 2 and 3) space without significant loss of visual fidelity. In experiments, we show FINOLA is more efficient than DCT and DWT.
> > >
> > > 2. *Self-Supervised Learning*: This property can be used to create mask prediction tasks for self-supervised learning, where the model learns effective representations by predicting masked regions using a simpler first-order autoregression (FINOLA).
> > >
> > > 3. *Computational Imaging*: The invariance property can potentially lead to new regularization techniques in inverse problems like FWI and CT. For instance, in FWI where we predict subsurface velocity map (represented as a 2D image) from seismic waveform data (represented as a 2D image too), we can apply FINOLA on both modalities (velocity map and seismic waveform) and explore their relationship in the compressed $\mathbf{q}$ space. This may reveal new relationship in the compressed space simpler than the relationship in the original space (modeled by partial differential equations). Similarly, in CT, enforcing invariance in the compressed $\mathbf{q}$ space could improve image reconstruction by reducing artifacts and noise.
> > >
> > > We believe that these additions will strengthen the paper and provide a clearer picture of the potential impact of FINOLA's invariance property.
> > >
> > > ---
> > > $\color{blue}{\textbf{[Question 1]:}}$
> > >
> > > **Latent Speeds (Abstract, Line 2): What does "latent speeds" refer to in the second line of the abstract?**
> > >
> > > To clarify, "latent speeds" refer to the eigenvalues in Equations (4) and (5). After diagonalization, each channel follows a finite approximation of the one-way wave equation:
> > >
> > > $\frac{\partial{\zeta}_k}{\partial{x}}=\lambda_k\frac{\partial{\zeta}_k}{\partial{y}}$
> > >
> > > where the eigenvalue $\lambda_k$ represents the wave speed. These eigenvalues are termed "latent speeds" because they are not explicitly used during model training but are implicitly derived in a post-training analysis.
> > >
> > > ---
> > > $\color{blue}{\textbf{[Question 2]:}}$
> > >
> > > **Optimality of FINOLA Solution (Eq. 2): Why might the solution of FINOLA in Equation 2 not be the optimal one? Isn’t it supposed to be unique?**
> > >
> > > *Optimality*
> > >
> > > FINOLA (Equation 1), expressed as $\Delta_x\mathbf{z}=\mathbf{A}\widehat{\mathbf{z}}$ and $\Delta_y\mathbf{z}=\mathbf{B}\widehat{\mathbf{z}}$, represents a special case of the more general Equation 2: $\Delta_x\mathbf{z}=\mathbf{AB}^{-1}\Delta_y\mathbf{z}$. Consequently, the solution space of FINOLA constitutes a subspace of the solutions to Equation 2.  If we assume that an image's feature space adheres to Equation 2, the optimal solution might lie outside this FINOLA subspace.
> > >
> > > This is supported by the introduction of Multi-Path FINOLA, which aggregates a series of FINOLA solutions (see Equation 3). This aggregated solution no longer satisfies Equation 1 but still satisfies Equation 2, effectively moving it outside the FINOLA subspace. Our experiments demonstrate the superior performance of Multi-Path FINOLA compared to standard FINOLA, providing evidence that the FINOLA solution is not always optimal.
> > >
> > > *Uniqueness*
> > >
> > > It's important to note that the solution to the one-way wave equation (Equation 2 or Equation 5, which decouples channels after diagonalization) is *not* unique. Any differentiable function of the form $f(\lambda x+y)$ satisfies $\frac{\partial{\zeta}}{\partial{x}}=\lambda\frac{\partial{\zeta}}{\partial{y}}$.

---

> > > > ### Author Response · Authors · 2024-11-24
> > > > **Response from authors - part 4**
> > > >
> > > > $\color{blue}{\textbf{[Question 3]:}}$
> > > >
> > > > **Definition of (Equation 3): What is $\widehat{\phi}$ in Equation 3? If it represents a normalization of $\phi$, how is this performed? Is it similar to the normalization for in Equation 1?**
> > > >
> > > > Yes, $\widehat{\phi}$ represents a normalized version of $\phi$. This normalization is performed identically to the normalization used in Equation 1: we subtract the mean and divide by the standard deviation.
> > > >
> > > > ---
> > > > $\color{blue}{\textbf{[Question 4]:}}$
> > > >
> > > > **Feature Map Resolution (Line 195): Is FINOLA unable to produce a feature map at a resolution of 1/8 of the original image?**
> > > >
> > > > FINOLA is capable of producing feature maps across a wide range of resolutions, including 1/8 of the original image size. As demonstrated in Table 1 (line 324), FINOLA effectively generates feature maps for all resolutions tested, from very low resolution (1/32 or 8x8) to full resolution (1 or 256x256). This includes the 1/8 resolution (32x32).
> > > >
> > > > In line 195, we focus on the higher resolutions (1/4, 1/2, and 1) because these pose greater computational challenges and require significantly more steps in the FINOLA process. However, this does not imply any limitation in FINOLA's ability to handle lower resolutions like 1/8.
> > > >
> > > > ---
> > > > $\color{blue}{\textbf{[Question 5]:}}$
> > > >
> > > > **Data Dependency of Learned $\mathbf{A}$ and $\mathbf{B}$: Are the learned matrices $\mathbf{A}$ and $\mathbf{B}$ data-dependent? If the test set changes, would the learned $\mathbf{A}$ and $\mathbf{B}$ no longer be applicable?**
> > > >
> > > > While matrices $\mathbf{A}$ and $\mathbf{B}$ ideally represent invariant spatial relationships across all images, in practice, we learn them from a specific training dataset. This learning process allows them to capture generalizable patterns and features from the training images.
> > > >
> > > > Despite being learned from a specific dataset, $\mathbf{A}$ and $\mathbf{B}$ can be effectively applied to any test set, even containing images unseen during training. This is because they encode underlying spatial relationships common to many images.
> > > >
> > > > For example, the exemplar images in Figure 2 were not part of the training or test set; they were downloaded from the web. This demonstrates that the learned matrices generalize well to new, unseen data. Therefore, even if the test set changes, the learned $\mathbf{A}$ and $\mathbf{B}$ remain applicable and do not need to be re-learned.

---

> > > > > ### Comment · Reviewer_9qTq · 2024-11-30
> > > > >
> > > > > I thank the authors for the detailed rebuttal. My concern regarding the 1.1, 1.2, and 1.5 is addressed by the authors' response. However, I still think section **Relaxing the FINOLA constraint through FINOLA series** needs to be justified by using results more than improved PSNR (Table 1) ( Is the improved quality the only metric authors used to detect optimality of the solutions?). Could the authors offer further clarification on this point?
> > > > > I agree with reviewer PCAc regarding the value of the paper. The proposed perspective is interesting, and the extensive experimental results are a strong aspect of the work.

---

> ### Author Response · Authors · 2024-11-30
> **Additional Justification for the FINOLA Series**
>
> Thank you for raising this important question. You're right to point out that relying solely on PSNR might not be sufficient to justify the FINOLA series. While improved PSNR is a key indicator, we also considered the impact on ***parameter efficiency***.
>
> FINOLA series (or multi-path FINOLA) not only improves PSNR with the same number of parameters but also achieves comparable or better PSNR with a significant reduction in the size of matrices $\mathbf{A}$ and $\mathbf{B}$. This reduction in parameters provides an important validation.
>
> For example, in Table 1, multi-path with 4 paths and 1024 channels outperforms single-path with 3072 channels across all resolutions. Importantly, multi-path with 1024 channels only has 1024x1024=1M parameters in matrices $\mathbf{A}$ or $\mathbf{B}$, which is 9 times less than the single-path with 3072 channels (requiring 3072x3072=9M parameters).
>
> Furthermore, Figure 5 shows that similar PSNR (between 25 and 26) can be achieved by using 16 paths and 256 channels, costing only 256x256=64K parameters in matrices $\mathbf{A}$ or $\mathbf{B}$ (144 times less than single-path with 3072).
>
> Therefore, relaxing the FINOLA constraint through FINOLA series offers a favorable trade-off: comparable or better image quality with significantly reduced the model complexity (achieved by reducing the size of matrices $\mathbf{A}$ and $\mathbf{B}$). This significantly reduces the number of one-way wave equations.
>
> In summary, while improved PSNR is a key result, the justification for FINOLA series also lies in its ability to achieve comparable performance with significantly reduced number of parameters. This highlights the potential of FINOLA series for efficient image representation.

---

### Meta-Review · Area_Chair_NTAN · 2024-12-27

**Metareview:**

The paper studies image representation. It develops an encoder-decoder pair with a special structure: the encoder maps the input image to a C dimensional feature vector q. Meanwhile, the decoder applies learned spatially invariant operations as follows: the vector q is placed at the center pixel, and then propagated outward via a learned difference equation (in feature space), in which increments are generated by applying learned matrices A and B to statistically normalized features. The paper observes that in this scheme, the matrices A and B are not only position-independent — they are in fact image-independent. This entire scheme is called FINOLA (first order norm + linear autoregression). The fact that these difference equations can be chosen in an image-independent fashion can be considered an approximate invariance associated with natural images — these images satisfy a common system of equations. The paper explores a multi-path variant which satisfies a one-way wave equation relating the horizontal and vertical increments in feature space. Experiments compare this image representation to classical transform coding (DCT/DWT/JPEG) and convolutional autoencoders, and apply the proposed representation to self-supervised learning.

On the positive side, the paper introduces a novel approach to autoencoding through a learned autoregressive process in feature space. This approach outperforms classical transform coding baselines. As the authors note, it is interesting that the spatial structure of images can be reproduced by applying a space invariant learned operation in feature space. At the same time, the paper frames its main contribution as discovering an invariance satisfied by natural images (rather than producing SOTA results on image representation); this contribution would be stronger with clearer intuitions for the nature of this invariance, i.e., what the matrices A and B are capturing. Reviewers produced a mixed evaluation of this core contribution, ranging from skepticism to enthusiasm. As noted by one of the reviewers, it is not clear if this particular encoder-decoder structure outperforms simpler baselines such as reshaping.

**Additional Comments On Reviewer Discussion:**

Reviewers were mixed in their initial evaluation of the paper. The paper provides a novel take on autoencoding, in which images are encoding via wave equations in feature space. The paper interprets the effectiveness of this method with a fixed pair (A,B) as a kind of invariance on images.

All reviewers expressed curiosity about the meaning of this invariance: does this reveal a fundamental property of natural images, or is it, as suggested by several reviewers, merely an artifact of the large number of degrees of freedom (as A, B are C x C matrices acting on a C dimensional feature space, and C can be chosen to be large). Reviewers also noted the unclear theoretical justification and interpretation for this observation.

Reviewers generally praised the experimental section, while also noting that the comparisons are mostly with baseline methods (DCT, DWT), rather than the state-of-the-art in autoencoding. There were also questions regarding other baselines (e.g., reviewer cywh asked comparison to a simpler encoding method based on reshaping q from a vector into a feature map, which achieves comparable performance).

---

### Decision · Program_Chairs · 2025-01-22

Reject